# High-Throughput Additive Manufacturing of Continuous Carbon Fiber-Reinforced Plastic by Multifilament

**DOI:** 10.3390/polym16050704

**Published:** 2024-03-05

**Authors:** Yiwen Tu, Yuegang Tan, Fan Zhang, Shulin Zou, Jun Zhang

**Affiliations:** School of Mechanical and Electronic Engineering, Wuhan University of Technology, 205 Luoshi Street, Hongshan District, Wuhan 430062, China; ywtu@whut.edu.cn (Y.T.); ygtan@whut.edu.cn (Y.T.); zsl940@whut.edu.cn (S.Z.); junzhang1128@whut.edu.cn (J.Z.)

**Keywords:** high-throughput, carbon fiber composite, robotic additive manufacturing, roller, laser

## Abstract

Additive manufacturing (or 3D printing) of continuous carbon fiber-reinforced plastics with fused deposition modeling is a burgeoning manufacturing method because of its potential as a powerful approach to produce lightweight, high strength and complex parts without the need for a mold. Nevertheless, it cannot manufacture parts rapidly due to low throughput. This paper proposes a high-throughput additive manufacturing of continuous carbon fiber-reinforced plastics by multifilament with reference to fiber tape placement. Three filaments were fed and compaction printed simultaneously by a robotic manufacturing system. The coupled thermal-mechanical model of the filament deformation during printing was developed to eliminate the initial interval between the filaments and improved mechanical properties. Furthermore, the mathematical relationship between filament deformation and printing parameters consisting of printing temperature, printing speed and roller pressure was proposed using response surface methodology with the line width as the response. The tensile tests demonstrate that the tensile properties of printed parts are positively correlated with the line width, but not infinitely improved. The maximum tensile strength and tensile modulus are 503.4 MPa and 83.11 Gpa, respectively, which are better than those obtained by traditional methods. Void fraction and scanning electron microscope images also reveal that the appropriate line width achieved by the reasonable printing parameters contributes to the high-throughput multifilament additive manufacturing of continuous carbon fiber-reinforced plastics. The comparison results indicate that the high-throughput multifilament additive manufacturing proposed in this paper can effectively improve the speed of continuous carbon fiber-reinforced plastics additive manufacturing without degrading the mechanical performance.

## 1. Introduction

Continuous carbon fiber-reinforced plastic (CCFRP) is widely used in aerospace, automobile manufacturing, petrochemical, and other industrial fields due to its higher strength-to-weight ratio and modulus compared with metals and alloys [1,2]. Additive manufacturing of CCFRP based on Fused Deposition Modeling (FDM) needs no mold and presents the characteristics of being low cost, lightweight, and high strength; it has gained increased attention within the academic and industrial fields in recent years [3].

Additive manufacturing takes layered manufacturing as the molding principle and integrates CNC, material science, computer-aided design and so on [4]. The continuous fibers can either be embedded into plastic through coaxial extrusion to create a prepreg filament suitable for manufacturing with FDM printers, or continuous fibers and plastic filaments can be fed into the nozzle of the printer simultaneously during the manufacturing process [5]. The plastic is heated to a molten state and extruded along with the carbon fibers. The relative movement of the nozzle and platform causes the plastic and carbon fibers to bond and cure on the platform layer by layer. 

Evidently, CCFRP involves plastic as the matrix and continuous carbon fiber as the reinforcement. The matrix is a base material that holds the carbon fibers in place for long-term performance and protects them from external forces such as corrosion, degradation, abrasion, etc. Matrix materials that are commonly used in the additive manufacturing of CCFRP are nylon (PA), polylactic acid (PLA), acrylonitrile-butadiene-styrene (ABS), and polyether-ether-ketone (PEEK) due to their thermoplastic standards and level of engineering [6,7,8].

To date, the research on additive manufacturing of CCFRP has emphasized the improvement of mechanical properties, techniques such as additive manufacturing in vacuumed conditions [9], heat treatment during additive manufacturing [10], annealing as a post-process [11], and compaction during additive manufacturing [12,13] have been proposed. The mechanical properties of additive manufactured CCFRP parts have been significantly improved. Nevertheless, low efficiency is one of the important reasons that hinder the industrial application of additive manufacturing of CCFRP. Research papers commonly report printing speeds below 10 mm/s [14,15,16,17]. Pappas et al. employed a screw to extrude PLA and then composited it with carbon fiber in situ, increasing the printing speed to 16 mm/s [18]. Li et al. melted the resin material by generating an eddy current in carbon fiber by microwave to effectively boost the printing speed [19]. Tu et al. used laser heating carbon fiber to melt resin so as to improve printing speed [20].

Different from additive manufacturing of CCFRP which employs a filament having a diameter less than 1 mm, automated tape laying (ATL) and automated fiber placement (AFP) utilizes tapes usually ranging from 6 to 75 mm in width [21]. The high-throughput technology lead to rapid manufacturing. Nevertheless, the challenge remains that the strip-based approach renders ATL and AFP less adept at tackling substantial curvature or sharp angles, impeding the fabrication of more complex configurations [22,23]. To increase the throughput and compensate for the low productivity due to the narrow tape, multi-feeding technology has been applied to AFP [24] and which can be also applied to the additive manufacturing of CCFRP [25].

This paper presents a pioneering exploration of an in-house developed robot-assisted high-throughput additive manufacturing of CCFRP using multifilament. To eliminate intervals between filaments and achieve better mechanical performance, the filament coupled thermal-mechanical model during printing was established and the mathematical relationship between line width and printing parameters was proposed. The tensile test, void fraction and scanning electron microscope (SEM) images reveal that the appropriate line width achieved by the reasonable printing parameters contributes to the mechanical performance of printed specimens. The comparative results fully explain the advantages of high-throughput multifilament additive manufacturing of CCFRP.

## 2. Materials and Methods

### 2.1. Materials

Polyacrylonitrile-based continuous carbon fiber 1k T300 of Zhongfu Shenying Carbon Fiber Co., Ltd. from Lianyungang, China was explored as the reinforcement, and 4032d transparent polylactic acid (PLA) from NatureWorks of Plymouth, MN, USA was used as the matrix plastic because of its good printability and recyclability. The CCFRP with a diameter of 0.5 mm (±0.05 mm) was prepared in self-designed prepreg equipment, as exhibited in Figure 1. The carbon fiber tows are drawn into the prepreg device by the winding roller, and the PLA filament is fed into the prepreg device and melted into liquid. The prepreg device contains five mixture pins which create a normal tension force between the fibers and the pin surfaces. This normal tension forces compression in the fibers over the pin surface to spread more and encourages melted PLA to infiltrate between the fiber bundles. The PLA-impregnated and -coated carbon fiber tows then enter the forming device and are compressed into 0.5 mm diameter CCFRP prepreg filaments, which are cooled and wound into rolls. The fiber volume fraction is about 19.6%, evaluated by:(1)$$Vf=\frac{k\cdot d^{2}}{D^{2}}$$
where Vf denotes the volume fractions of carbon fiber; k and d represent the specification (1k) and diameter (7 μm) of single tow of carbon fiber, respectively; D signifies the diameter of the CCFRP.

### 2.2. Additive Manufacturing System

#### 2.2.1. Hardware Setup

This in-house developed printing system encompasses a six-axis robot, a two-axis positioner as platform, three filament feeders, a multifilament print head and a PC as the host controller, as depicted in Figure 2. The rated load at the end joint of the robot (SR20-1700 from STEP, Shanghai, China) is 20 kg. The print head is installed on the end joint of the robot and the hot roller is located on the Z-axis extension line of the end joint. Above the print head are filament cutters to handle discontinuous paths [26]. The guiding device can restrain the CCFRP to guarantee that it enters the laser irradiation area and is arranged at fixed intervals. The hot roller provides the pressure required for compaction printing, and the printing pressure can be measured and adjusted using the pressure sensor and linear motor. Considering the high absorption of laser energy by carbon fibers, a solid-state pulse laser with a rated power of 5.5 W is adopted to heat the CCFRP filament rapidly, as introduced in previous work [20]. An infrared temperature sensor is used to detect the temperature of the CCFRP and then the laser power is controlled by a pulse width modulation (PWM) signal based on the measured temperature.

#### 2.2.2. Software and Control System

As shown in Figure 3, the information flow commences with the CAD model of the part. The model is converted into an initial path file in the slicer. The predetermined print parameters are then funneled into path file. The final G-code file is produced after interpolation using specialized motion planning algorithms.

EtherCAT technology is used for synchronous and coordinated control of the robots and the print head. Simple Open EtherCAT Master Library (SOME) is used as an EtherCAT master on a Linux computer. The robot controller and print controller are configured as two EtherCAT slaves. Robot controllers are used to control the motion of six-axis robots and two-axis positioners. The print controller is used to control the cutters, heaters, hot roller and feeders.

#### 2.2.3. Prototyping Principle

Three CCFRP filaments are fed into the guide device from dedicated feeders, which reduce the distance between the filaments and arrange them at fixed intervals through curved channels in the guide device, as shown in Figure 4a. The filaments are heated to a molten state by laser and then compacted by the hot roller. The CCFRP filaments are deformed by the combination of compression and high temperature from the hot roller. As shown in Figure 4b, The cross-section of the CCFRP filaments changes from a circle with radius R to a rectangle with width b and h.

There are initial intervals D (1.5 mm) due to the limitation of the guiding device between multifilament before formation. The printing parameters must be reasonably well defined to allow sufficient deformation of the CCFRP to eliminate the initial intervals and avoid porosity resulting in performance degradation.

### 2.3. Filament Deformation Coupled Thermal-Mechanical Model

CCFRP filament deforms and flows under thermal coupling during the additive manufacturing process. It can be treated as an extrusion flow of resin between parallel plates if the permeability of the resin between the carbon fibers is ignored. Assuming that the resin is a Newtonian fluid [27], squeezing the flow between parallel plates does not change the volume of the resin. The formula is:(2)V=A(0)·H(0)=A(t)·H(t)

As shown in the Figure 5, V is the volume of resin, $A\left(0\right)$ and $H\left(0\right)$ are the initial contact area and height before squeezed, and $A\left(t\right)$ and $H\left(t\right)$ are the area and height at a certain moment.

The combination of the power-law fluid proposed by Scott [28] and the theoretical model proposed by Leider [29] et al. leads to an equation for the squeezed flow between parallel plates, which is similar to the automated fiber placement manufacturing [30].
(3)$$F=m\pi \frac{{\left(2+1/n\right)}^{n}{\left(-H^{\prime }\right)}^{n}R^{3+n}}{2^{n}(3+n)H^{1+2n}}$$
where m and n are the model coefficients of the power-law fluid, H is the height of the fluid, $H^{\prime }$ is the first order derivative and R is the radius of the contact area.

If the resin can be assumed to be a Newtonian fluid at a constant temperature and speed during printing, the above equation can be simplified to:(4)$$F=\frac{3\pi \mu (-H^{\prime })R^{4}}{8H^{3}}$$

The volume of the fluid in Figure 5 is $V=AH=\pi R^{2}H$, which is a fixed constant when the wire properties are constant. This is obtained by substituting it into Equation (4) and integrating both sides simultaneously:(5)$$\int_{0}^{t}Fdt=-\frac{3\mu }{8\pi }V^{2}\int_{H(0)}^{H(t)}\frac{1}{H^{5}}dH$$

During the printing process, the pressure hot roller on the filament is related to the depth of contact, as shown in Figure 6.

As shown in Figure 7, the length of the CCFRP filament is L, the printing speed is v, and the pressure of the hot roller on the CCFRP is $F(t_{1})$ at position 1, which changes to $F(t_{1}+2L/v)$ at position 2. Position 1 is the moment when the hot roller just touches the filament section and position 2 is the end moment.

As shown in the Figure 8, in the above printing process, the pressure at the moment $t_{0}$ is $F(t_{0})$ and the contact area is $s_{1}$, and at the moment $t_{0}+L/v$ is $F(t_{0}+L/v)$ and the contact area is $s_{2}$.

The pressures at the moments $t_{0}$ and $t_{0}+L/v$ are, respectively:(6)$$\left\{ \begin{array}{l} F(t_{0})=\int_{0}^{s_{1}}dF\\ F(t_{0}+L/v)=\int_{0}^{s_{2}}dF \end{array}\right.$$
where $s_{1}+s_{2}=L$, the complete compaction printing process can be expressed as:(7)$$\int_{0}^{t}Fdt=\int_{0}^{\frac{2L}{v}}F(t)dt=\int_{0}^{\frac{L}{v}}F(t)dt+\int_{\frac{L}{v}}^{\frac{2L}{v}}F(t)dt$$

This simplifies to:(8)$$\int_{0}^{t}Fdt=F\frac{L}{v}$$

It can be derived from Equations (5) and (8):(9)$$F\frac{L}{v}=\frac{3\mu }{32\pi }V^{2}\left(H^{-4}(t)-H^{-4}(0)\right)$$

CCFRP is susceptible to plastic deformation at high temperatures and pressures, whereas the hot roller can be treated as a rigid body. There is therefore a geometric relationship between L and the roller radius and depth of contact:(10)$$L=\sqrt{R^{2}-{\left[R-H_{0}+H(t)\right]}^{2}}$$

As shown in the Figure 9, $H_{0}$ is the initial height of the filament, H(t) is the height after compaction and R is the radius of the hot roller.

It can be concluded that without considering the fiber volume content and its distribution characteristics, the line width as well as the layer thickness of the formed CCFRP is only related to the printing temperature, the printing speed and the pressure of the hot roller, as shown in the Table 1.

### 2.4. Mathematical Model of Filament Deformation and Printing Parameters

For further quantitative analysis of the above printing parameters in relation to the deformation of CCFRP filaments during printing, a series of experiments based on response surface methodology (RSM) were carried out, as shown in Figure 10 and Table 2.

The layer thickness is obtained by averaging several measurements, while the line widths are calculated following the principle of volume invariance.
(11)$$\pi d^{2}=4bh+\pi h^{2}$$
where d is the filament diameter of the CCFRP, and b and h are the line width and layer thickness after compaction printing, respectively, as shown in Figure 11.

The linear fit model was selected as the response expression model after synthesizing the comparisons. The line width has the following relationship with printing parameters:(12)b=−0.157579+0.008786⋅t−0.049864⋅v+0.15444⋅p
where b is the line width and t, v and p are the printing temperature, printing speed and hot roller pressure, respectively. The absolute value of each coefficient indicates the extent to which the parameter affects line width, and the signs indicate positive and negative correlations. The mathematical model fitting results agree with the analytical results of the coupled thermal-mechanical model. In theory, the best print quality is achieved when the line width b is equal to the multifilament initial intervals D (1.5 mm).

### 2.5. Printing Experiments and Tensile Properties Test

Several groups of unidirectional CCFRP specimens were printed with different theoretical line width and tensile properties were tested to find out optimal print parameters. All specimens were printed with linear paths, as illuminated in Figure 12. Each path is 400 mm long and consists of three filaments and each layer contains two paths for a total of six filaments. After each path is printed done, the carbon fiber filaments are sheared off and the print head returns to the start point to print the next path. Printed as a result of various combinations of parameters, all specimens consist of 15 layers with thicknesses ranging from 1.51 mm to 2.63 mm and width ranging from 8.12 mm to 12.16 mm.

The dimensions of the tensile test specimen were theoretically 120 × 10 × 2 mm (length × width × thickness) in accordance with GB/T 1447-2005 [31]. Each printed specimen is cut into three tensile specimens of 120 mm length and varying widths and thicknesses.

The mechanical properties test was performed using a universal testing machine (WDW-20KN, Sanfeng, Changzhou, China). The tensile test was conducted at a loading speed of 2 mm/min, and the tensile strain was obtained by attaching an electronic extensometer to the specimen.

### 2.6. Void Determination and Scanning Electron Microscopy

Precision electronic balance (AL240, Mettler Toledo) was employed to measure specimen mass. The void fraction of the specimens was measured by the Archimedes method. Specifically, $m_{1}$ denotes the mass of the specimen in the air after drying; $m_{2}$ represents the mass of the specimen after soaking in water for 72 h; $m_{3}$ signifies the mass of the water-saturated specimen in water. The void fraction (ε) can be calculated by:(13)$$\varepsilon =\frac{m_{2}-m_{1}}{m_{2}-m_{3}}$$

In addition, the fracture surfaces of the tensile specimens were observed using scanning electron microscopy (SEM) (JSM-IT300, JEOL from Tokyo, Japan) to investigate the void distribution and bonding surface.

## 3. Results and Discussion

### 3.1. Void Fraction and Cross-Sectional Observation

The void fraction of the all-printed specimens was determined using the Archimedes method described in Section 2.6. Figure 13 demonstrates the void fraction of the specimens with different line widths. The void fraction of the specimens is negatively correlated with the line width. Furthermore, the changing rate of void fraction is also negatively correlated with the line width. It tends to rapidly decrease with the increasing line width when it is close to 1.5 mm and then decreases gently.

Typical examples of cross-sectional SEM images of printed specimen failure surface after the tensile tests are illustrated in Figure 14, which explain the formation and variation of voids. The filament deformation is poor to fill the initial intervals when the line width is much less than 1.5 mm, resulting in a large gap between the filaments, which is the reason for the high void fraction.

As the line width approaches 1.5 mm, the void gradually decreases but still remains. This could be because the filament sides are semicircular after being pressed and deformed, resulting in incomplete contact, as shown in Figure 15a.

Increasing the line width further means increasing the printing temperature, reducing the printing speed or increasing the roller pressure, which makes the filaments more susceptible to deformation and compresses the voids between the filaments as analyzed in Section 2.3 and Section 2.4. When the roller pressure is high enough, the voids between the filaments would almost disappear, leaving only the pores inside the filaments, just as shown in Figure 15b.

### 3.2. Tensile Properties

The tensile properties of multifilament additive manufacturing of CCFRP should be systematically characterized with different line widths which are the integrated response of multiple key printing parameters to evaluate the feasibility of CCFRP high-throughput prototyping. The effect of line width on the tensile properties of specimens is depicted in Figure 16a. The longitudinal tensile strength and tensile modulus are generally positively correlated with the line width. Furthermore, the tensile strength is approximately linear with line width when it is less than the multifilament initial intervals (D = 1.5 mm). As the line width approaches 1.5 mm, the tensile strength change rate tends to increase slightly while the tensile modulus change rate decreases significantly. The rate of change of tensile strength and modulus tends to zero when the line width exceeds 1.8 mm. Figure 16b shows the representative stress vs. strain curves of specimen with 1.785 mm line width which was printed at 210 °C, 2.5 mm/s printing speed and 4.5 N (for three filaments) printing pressure.

The minimum tensile strength and tensile modulus are 319.06 MPa and 66.9 GPa, respectively and the maximum are 503.4 MPa and 83.11 Gpa, respectively. The presence of voids within the printed carbon fiber composite specimens was discovered to exert substantial negative impacts on mechanical performance [31]. The narrow line width specimens are more susceptible to shear failure due to the high void fraction. As shown in Figure 17a, the specimen with a line width of 1.189 mm fractured in cross section after splitting from the center with the tensile force of 9.318 kN. In contrast, specimens with wider line widths are more prone to brittle fracture due to stretching, as shown in Figure 17b.

Actually, even if the printing speed is increased by more than 20 mm/s, since the temperature cannot be increased indefinitely, excellent mechanical properties can still be maintained as long as enough roller pressure is adapted.

### 3.3. Comparison and Printed Parts Demonstration

The comparison of the changes in tensile strength of additively manufactured CCFRP parts concerning fiber volume fraction [3,32,33,34,35,36,37,38,39,40,41] is provided in Figure 18a. Evidently, the CCFRP specimens produced in this study exhibit significantly enhanced tensile properties at certain fiber volume fractions, particularly when compared to traditional printing processes. On the other hand, the comparison of the filament throughput is also shown. Given the differences in printing processes and filament specifications, the print path length per second is used as the standard for comparison [14,15,16,18,19,25,42,43,44,45,46,47,48], as shown in Figure 18b.

In Figure 19, several parts produced by high-throughput multifilament additive manufacturing are presented. Square and T-shaped parts are 4.2 mm thick and each layer contains only one complete path. The number of paths per layer for the notched and rectangular parts are two and four, respectively. Print times for the parts in this paper are much shorter than traditional single filament additive manufacturing of CCFRP.

## 4. Conclusions

The high-throughput multifilament 3D printing process of continuous carbon fiber-reinforced plastics (CCFRP) is proposed in this paper. Three CCFRP filaments were fed and compaction printed simultaneously by a robotic manufacturing system.

The technology’s hardware encompasses a manufacturing system that includes a two-axis positioner as platform, a six-axis robot, a specialized print head, and the EtherCAT control system.

The filaments are arranged at fixed intervals by guide device and then heated and compacted by a laser and hot roller, respectively. The coupled thermal-mechanical model of the filament deformation during printing was developed to eliminate the initial interval between the filaments. The mathematical relationship between filament deformation and printing parameters consisting of printing temperature, printing speed and roller pressure was proposed using response surface methodology with the line width as the response.

Furthermore, the void fraction and tensile properties of printed specimens are closely related to line width and the specific relationship is related to the initial filament intervals. The results show that the appropriate line width achieved by the reasonable printing parameters can fully highlight the mechanical properties of the printed specimens. The maximum tensile strength and tensile modulus are 503.4 MPa and 83.11 GPa, respectively, which better than the traditional printing method. Several printed parts of different shapes demonstrate the stability and efficiency of high-throughput multifilament additive manufacturing of CCFRP.

In summary, the high-throughput multifilament additive manufacturing proposed in this paper can effectively improve the speed of CCFRP additive manufacturing without degrading the mechanical performance, which provides technical and theoretical support for the industrialized application of CCFRP rapid additive manufacturing.

## Figures and Tables

**Figure 1 polymers-16-00704-f001:**
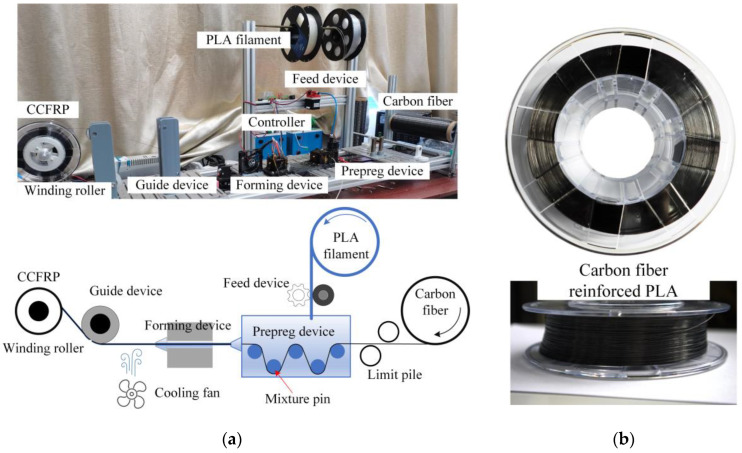
Continuous carbon fiber-reinforced plastics: (**a**) prepreg equipment; (**b**) carbon fiber-reinforced PLA.

**Figure 2 polymers-16-00704-f002:**
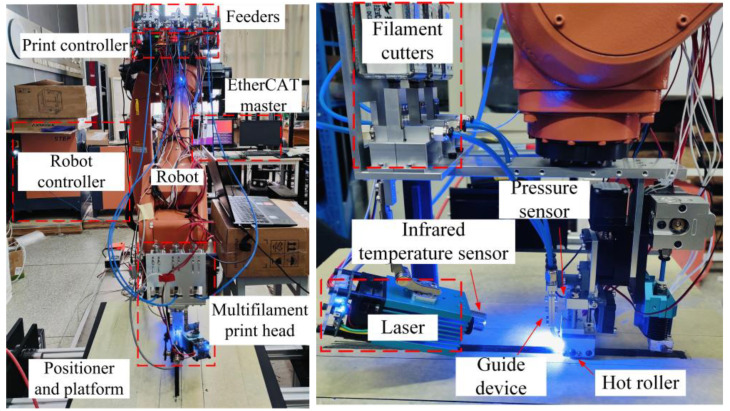
Additive manufacturing system and multifilament print head.

**Figure 3 polymers-16-00704-f003:**
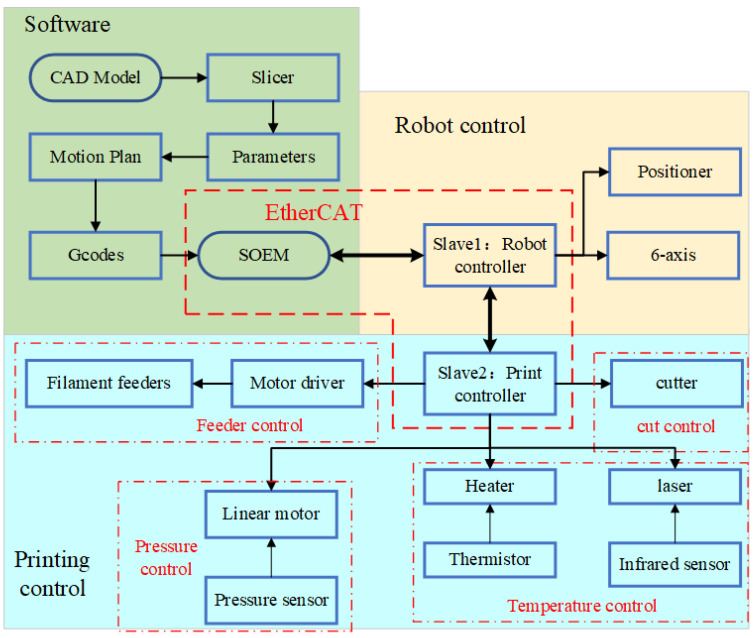
Control system of high-throughput multifilament additive manufacturing.

**Figure 4 polymers-16-00704-f004:**
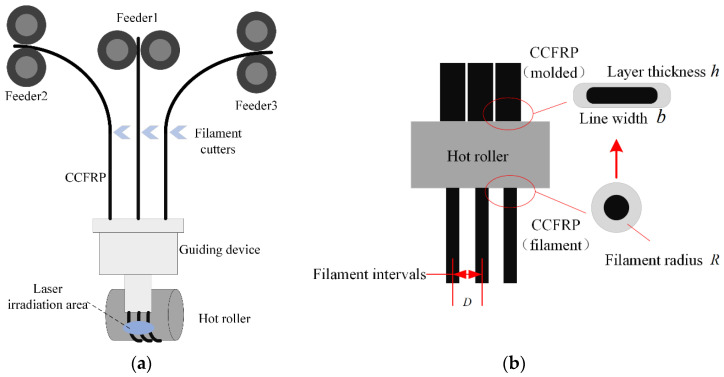
High-throughput multifilament additive manufacturing of CCFRP: (**a**) printing process; (**b**) filament intervals and deformation.

**Figure 5 polymers-16-00704-f005:**
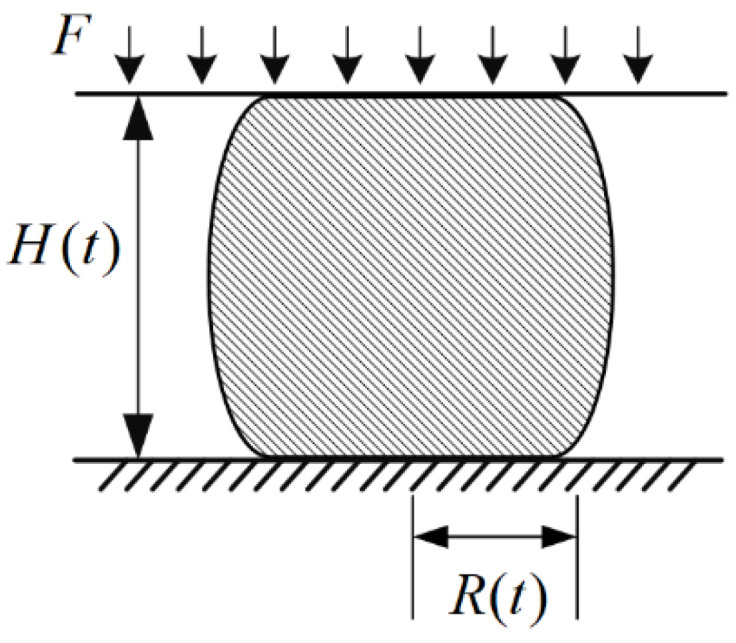
Extrusion flow of resin between parallel plates.

**Figure 6 polymers-16-00704-f006:**
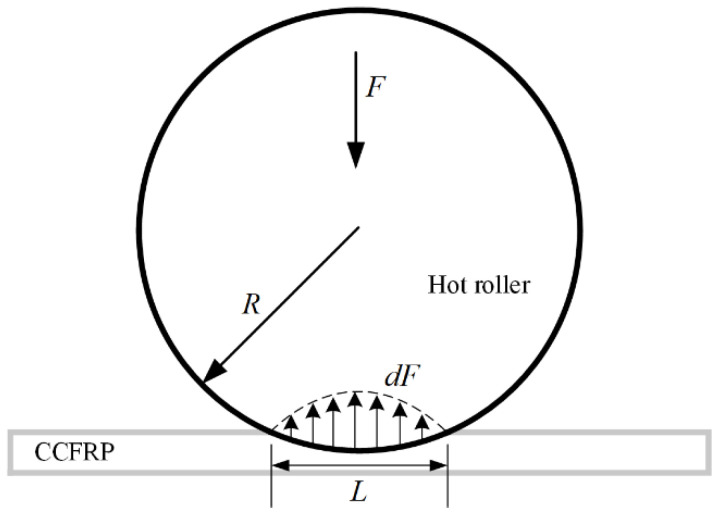
The contact of hot rollers with CCFRP.

**Figure 7 polymers-16-00704-f007:**
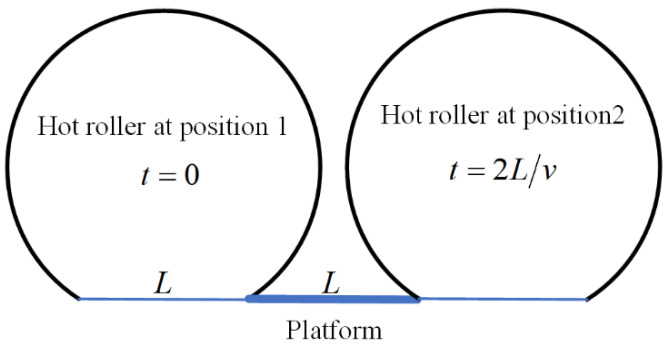
Before and after hot roller contact with CCFRP section.

**Figure 8 polymers-16-00704-f008:**
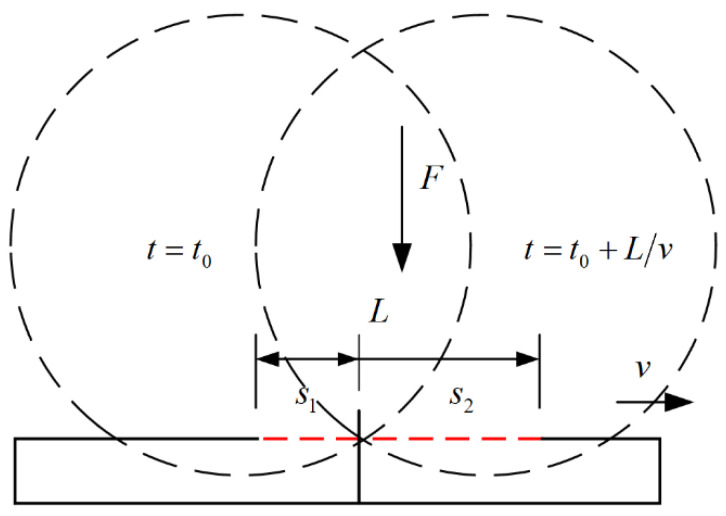
Contact between hot roller and CCFRP at $t_{0}$ and $t_{0}+L/v$.

**Figure 9 polymers-16-00704-f009:**
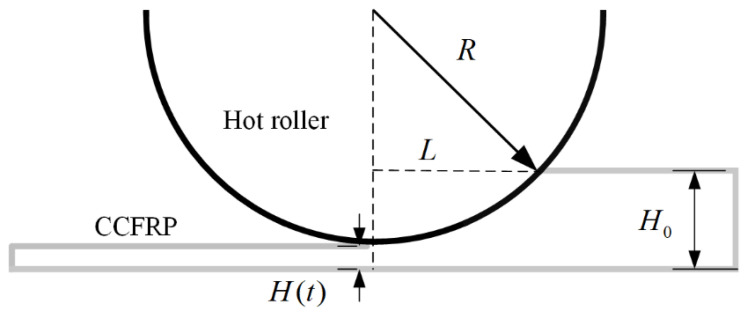
Length of hot roller contacting CCFRP.

**Figure 10 polymers-16-00704-f010:**
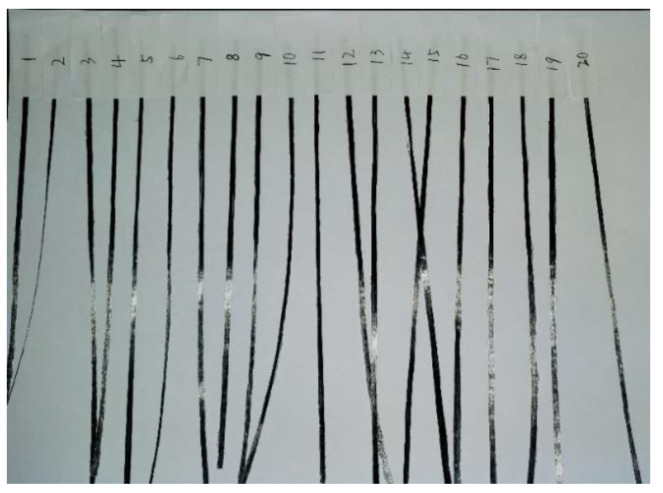
Line widths for different print parameters.

**Figure 11 polymers-16-00704-f011:**
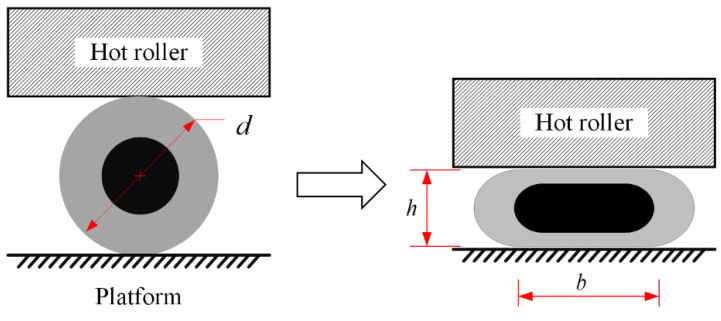
Dimensions of cross-section before and after filament deformation.

**Figure 12 polymers-16-00704-f012:**
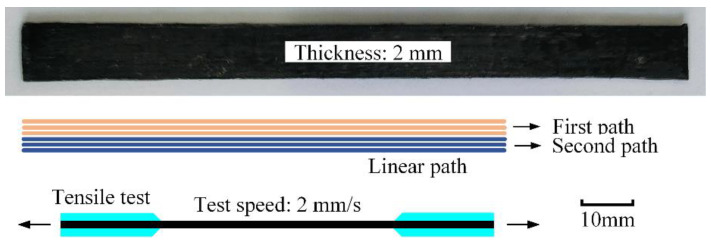
Specimen size, paths and test principle.

**Figure 13 polymers-16-00704-f013:**
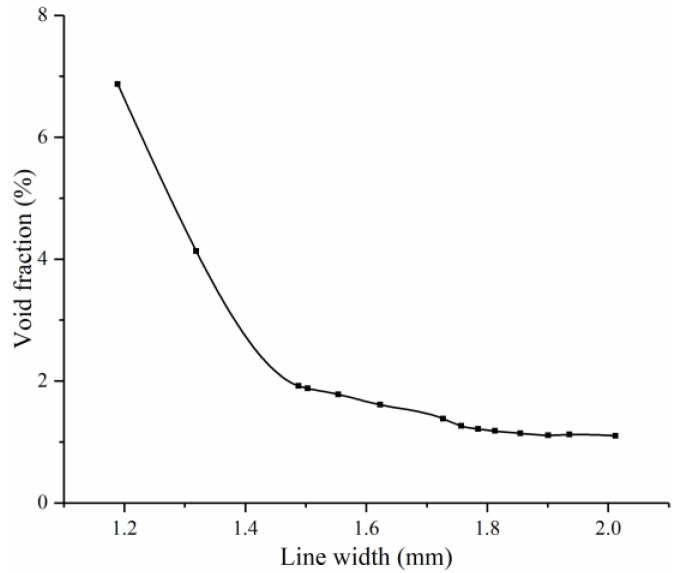
Specimen void fraction with different line width.

**Figure 14 polymers-16-00704-f014:**
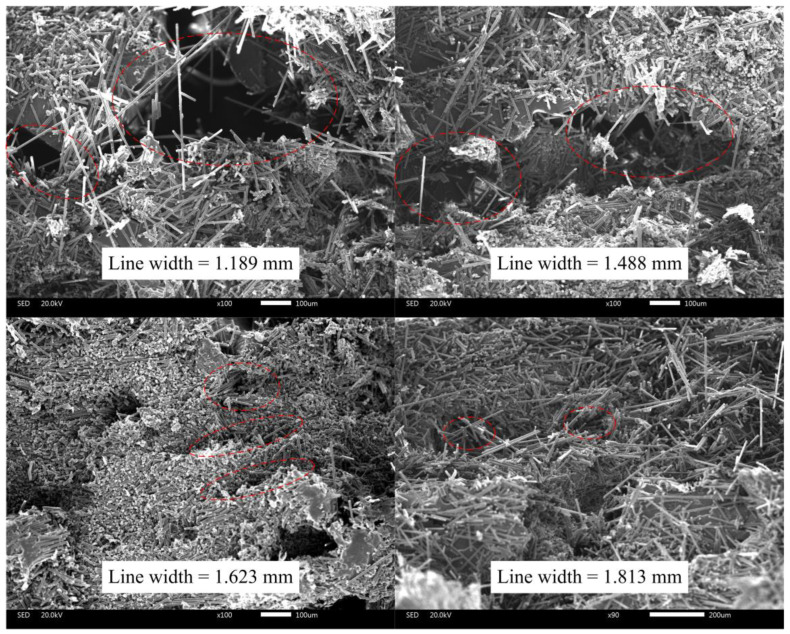
Cross-sectional SEM images of specimens with different line widths, and the obvious voids are circled with red dotted lines.

**Figure 15 polymers-16-00704-f015:**
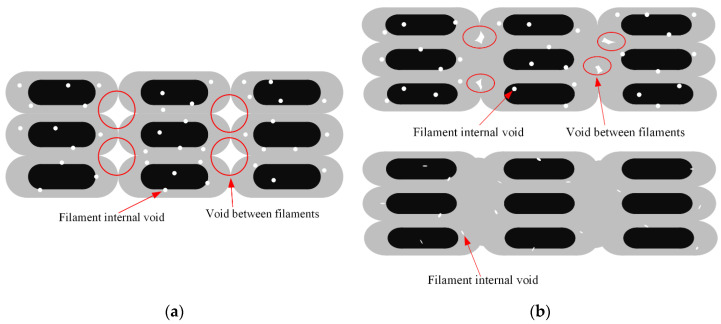
Void between filaments: (**a**) caused by incomplete contact; (**b**) poor void fraction due to high roller pressure.

**Figure 16 polymers-16-00704-f016:**
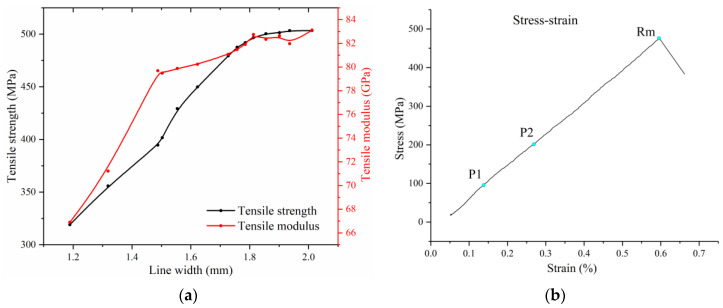
Tensile properties of printed specimens: (**a**) tensile strength and modules with different line widths; (**b**) stress vs. strain curves of specimen with 1.785 mm line width.

**Figure 17 polymers-16-00704-f017:**
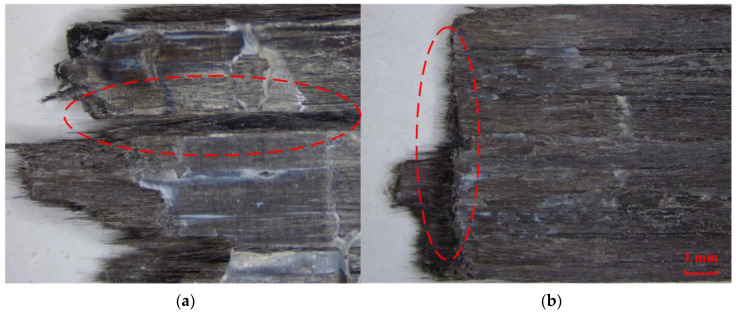
Failed specimens: (**a**) shear and tensile failure; (**b**) tensile failure. The failure surfaces are circled with red dotted lines.

**Figure 18 polymers-16-00704-f018:**
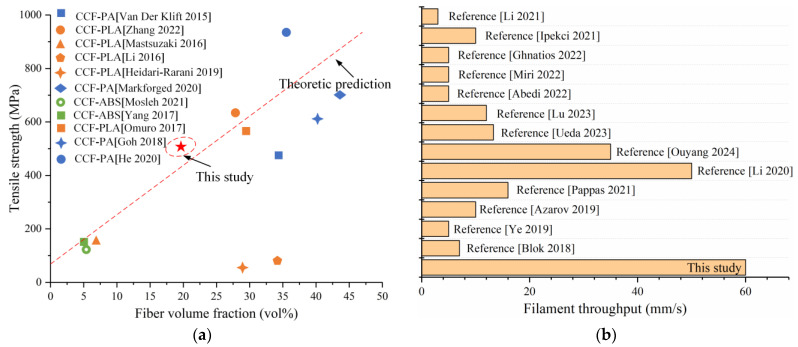
Comparison of printed parts: (**a**) tensile strength [3,32,33,34,35,36,37,38,39,40,41]; (**b**) filament throughput [14,15,16,18,19,25,42,43,44,45,46,47,48].

**Figure 19 polymers-16-00704-f019:**
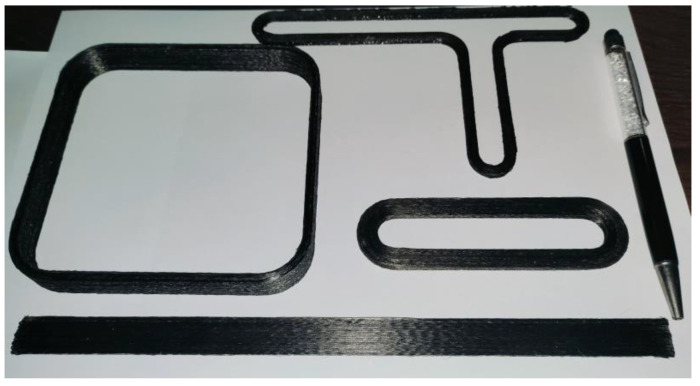
High-throughput multifilament additive manufactured parts of CCFRP.

**Table 1 polymers-16-00704-t001:** Effect of printing parameters on deformation of CCFRP.

Parameters	Trends	Mechanization	Line Width	Layer Thickness
Printing temperature	Up	Reduces resin viscosity	Widen	Reduction
Printing speed	Up	Reduced compaction time	Narrow	Increase
Pressure	Up	Improved compaction	Widen	Reduction

**Table 2 polymers-16-00704-t002:** Three-factor four-level experimental results of CCFRP parameters and deformation.

No.	Printing Temperature (°C)	Printing Speed (mm/s)	Pressure (N)	Layer Thickness (mm)	Line Width (mm)
1	220	10	0.5	0.148	1.327
2	220	10	2	0.126	1.554
3	200	5	2	0.1210	1.623
4	190	2.5	0.5	0.133	1.478
5	190	10	2	0.139	1.409
6	220	5	1	0.114	1.727
7	200	5	2	0.123	1.602
8	190	2.5	1.5	0.122	1.615
9	200	10	1	0.147	1.338
10	210	7.5	0.5	0.149	1.319
11	220	5	1	0.112	1.757
12	210	2.5	0.5	0.119	1.65
13	200	10	1	0.153	1.283
14	210	2.5	1.5	0.110	1.785
15	210	7.5	1.5	0.132	1.483
16	220	2.5	2	0.098	2.012
17	200	5	1	0.132	1.488
18	210	7.5	1.5	0.131	1.503
19	190	7.5	1.5	0.145	1.357
20	190	7.5	0.5	0.165	1.189

## Data Availability

Data are contained within the article.

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
