# Peer review of "High-Throughput Additive Manufacturing of Continuous Carbon Fiber-Reinforced Plastic by Multifilament"

_polymers, 2024, doi:10.3390/polym16050704_

Round 1

Reviewer 1 Report

Comments and Suggestions for Authors

The research presented in the article concerns an automated high-throughput system for obtaining fiber optic tape for additive technology. This process enables faster fiber processing while achieving better mechanical parameters. The work is very innovative and fits into new trends in high-throughput research. The presented system has many development possibilities. It will be difficult for other scientists to conduct research using this type of solution because it requires the creation of a unique tool. However, the presented process concept is new. I recommend that the work be accepted with some corrections:

1) The introduction, although it contains 21 references, is too short and should be expanded.

2) It is recommended to remove abbreviations from the abstract, the authors also abuse the term 3D printing throughout the text, which is a jargon term. The term additive technologies should be used interchangeably.

3) Sometimes there is a space between the values and units in the text and sometimes there is not, please enter the correct notation

4)Please standardize the references in the text, e.g. item [26], section 3.2.

5) Please check and correct the formatting of the literature in the bibiography

6) For readability, enlarge diagram 3 using the entire width of the text.

7) Reduce the distance between rows in tables - too much empty space

8) Improve the font quality in Figure 19 - unreadable

Reviewer 2 Report

Comments and Suggestions for Authors

1.       Equations require reference.

2.       How is the fracture surface in Figure 14 created?

3.       It is better to merge figures 15 and 16 together.

4.       It is better to compare and verify schematic figures 15 and 16 with SEM images.

Reviewer 3 Report

Comments and Suggestions for Authors

1.      The title is misleading as one would expect an in-depth analysis of multifilament printing. However, the authors mention the following paragraph: “This paper presents a pioneering exploration of an in-house developed robot-assisted high-performance 3D printing of multifilament using CCFRP. To eliminate gaps between filaments and achieve better mechanical performance,…’ The authors should modify the article's title, based on the previous paragraph.

2.      Define PLA

Materials and methods

3.      What is the diameter of 1k T300 polyacrylonitrile-based continuous carbon fiber?

4.      The authors have to indicate why to use polyacrylonitrile with PLA, are they compatible? Why try carbon fiber printing with a biodegradable polymer?

5.      The authors should explain the PLA in more detail. PLA 4032d is pellets; the authors note it is a filament in Figure 1a. Authors should review this section.

6.      The authors have to describe the manufacturing process of the CCFRP. Wouldn't that be a CCFRPLA? It is not clear how carbon fiber is integrated into PLA. The authors must post an image that outlines this process.

7.      The question arises again: is PLA the best polymer to integrate CCF?

2.2.3 Prototyping principle

8.      Authors must complement this section.

9.      The authors mention 'The filaments are heated to a molten state by laser and then compacted by the hot roller'.

10.    What is the effect of the laser on PLA? The authors must indicate the temperature that the laser radiates on the filament. Table 2 indicates the temperature, but how did they measure it and how is it controlled?

11.   PLA is a thermoplastic. Why use laser?

2.5 Printing experiments and tensile property testing.

This section has to be thoroughly reviewed by the authors.

12.   The authors indicated that the CCFRP filaments have a diameter of 5 mm, but in this section they indicate that ‘The dimensions of the tensile test sample were 120 × 10 × 2 mm (length × width × thickness), consisting of 3 filaments and Each layer contains two 214 paths for a total of 6 strands…’ The math doesn't add up, especially considering that a roll passes over the filaments, which would reduce the thickness even more.

13.   In the section on printing experiments and tensile property tests, the authors mention GB/T 3354-1999. This standard is obsolete. Why did they use an obsolete standard from 1999 for work carried out in 2023? What are the justifications for using the 1999 standard instead of a more recent (2014) standard?

14.   The authors should provide more detail about the manufacturing of the specimens. What is printing speed? What happens to the fiber when the print head has to return? Does it distort?

15.   Figure 14 shows SEM micrographs, but it does not indicate whether it is the failure surface (after the tensile test) or a micrograph prior to the mechanical test. The authors have to give more details to understand the micrographs. In post-mortem specimens, microcavitations are seen due to the detachment of the reinforcement and matrix. Considering that this printed specimen is a laminated-type, the failure should be due to delamination, which is why the micrographs are not well understood. If it is necessary to analyze the specimens's interior (without distortion due to mechanical testing), tomography or cryogenic cutting tests may be useful.

16.   Figure 16 is an image that speculates or shows a hypothesis. The authors cannot say that this really happens. Authors are recommended to provide results that support the paragraph and Figure 16, or modify the text, arguing that it is just an assumption.

17.   Add scale to Figure 18.

18.   Observing the high mechanical properties of CCFRP is curious, especially considering that the composite matrix is PLA. The authors must add the representative stress vs strain curves to know the mechanical behavior. It is not valid to just place a graph with points (Figure 17).

19.   Figure 20 is not necessary for this research article.

Round 2

Reviewer 2 Report

Comments and Suggestions for Authors

Thanks to the dear authors.

Manuscripts are acceptable.

Reviewer 3 Report

Comments and Suggestions for Authors

The authors responded the questions and suggestions